# Regulation of p53 Function by Formation of Non-Nuclear Heterologous Protein Complexes

**DOI:** 10.3390/biom12020327

**Published:** 2022-02-18

**Authors:** Lev Zavileyskiy, Victoria Bunik

**Affiliations:** 1Faculty of Bioengineering and Bioinformatics, Lomonosov Moscow State University, 119991 Moscow, Russia; zavilev@fbb.msu.ru; 2Department of Biokinetics, Belozersky Institute of Physico-Chemical Biology, Lomonosov Moscow State University, 119991 Moscow, Russia; 3Department of Biochemistry, Sechenov University, 119991 Moscow, Russia

**Keywords:** intracellular localization of p53, heterologous non-nuclear complexes of p53, p53 sequestration, p53 trafficking, thiol/disulfide-dependent regulation of p53 functions

## Abstract

A transcription factor p53 is activated upon cellular exposure to endogenous and exogenous stresses, triggering either homeostatic correction or cell death. Depending on the stress level, often measurable as DNA damage, the dual outcome is supported by p53 binding to a number of regulatory and metabolic proteins. Apart from the nucleus, p53 localizes to mitochondria, endoplasmic reticulum and cytosol. We consider non-nuclear heterologous protein complexes of p53, their structural determinants, regulatory post-translational modifications and the role in intricate p53 functions. The p53 heterologous complexes regulate the folding, trafficking and/or action of interacting partners in cellular compartments. Some of them mainly sequester p53 (HSP proteins, G6PD, LONP1) or its partners (RRM2B, PRKN) in specific locations. Formation of other complexes (with ATP2A2, ATP5PO, BAX, BCL2L1, CHCHD4, PPIF, POLG, SOD2, SSBP1, TFAM) depends on p53 upregulation according to the stress level. The p53 complexes with SIRT2, MUL1, USP7, TXN, PIN1 and PPIF control regulation of p53 function through post-translational modifications, such as lysine acetylation or ubiquitination, cysteine/cystine redox transformation and peptidyl-prolyl *cis-trans* isomerization. Redox sensitivity of p53 functions is supported by (i) thioredoxin-dependent reduction of p53 disulfides, (ii) inhibition of the thioredoxin-dependent deoxyribonucleotide synthesis by p53 binding to RRM2B and (iii) changed intracellular distribution of p53 through its oxidation by CHCHD4 in the mitochondrial intermembrane space. Increasing knowledge on the structure, function and (patho)physiological significance of the p53 heterologous complexes will enable a fine tuning of the settings-dependent p53 programs, using small molecule regulators of specific protein–protein interactions of p53.

## 1. Introduction

Protein p53 is a well-known transcription factor activated in stressed cells, such as cells deprived of oxygen or growth factors [1,2]. Depending on the ensuing level of DNA damage, p53 triggers either correction programs or cell death [3]. Although p53 is most studied in the context of tumor suppression and genome stability, these functions rely on the p53 “guardian” role, i.e., it addressing the homeostatic perturbations in normal cells [4,5]. The major and most studied mechanism of the p53 action is its binding to DNA, resulting in the activation, or, less often, inhibition of transcription of the p53 target genes [6,7]. The DNA-binding function is well known to be regulated by the interaction of p53 with a number of nuclear proteins [8,9]. In view of co-localization of these complexes with cellular genome, their role in the regulation of p53 transcriptional function is more obvious than the role of non-nuclear complexes. Owing to this, functional significance and structural features of the former rather than the latter have been characterized in a number of independent studies. Based on the molecular mechanisms of p53 interactions with nuclear targets, small molecule regulators to block or promote such interactions are designed, opening new perspectives for pharmacological regulation of the p53- or its mutant-dependent responses [10,11,12,13].

In addition to the nucleus, p53 is also found in other cellular compartments, such as mitochondria, endoplasmic reticulum and cytosol [14]. By sequestering the protein from nuclear accumulation [15], or dynamically changing the p53 distribution inside the cells [16], these non-nuclear localizations may regulate both the transcriptional and non-nuclear functions of p53 [10]. The p53 trafficking between the different cellular compartments, its molecular mechanisms and biological significance represent important aspects of p53 action. The aim of this review is to summarize the current information on the heterologous complexes of p53 with non-nuclear proteins and the role of such complexes in supporting a variety of biological functions of p53. 

## 2. Structural Determinants of Homo- and Heterologous Protein Complexes of p53

Different regions of p53 sequence fulfill specific functions in the p53 interactions with its targets. DNA-binding domain (DBD) is the major domain involved in transcriptional functions of p53 and their impairments upon p53 mutations in cancer (Figure 1) [17]. The domain structure is stabilized by Zn^2+^ ion bound to cysteine residues of the domain shown in Figure 2. Zn^2+^-free p53 at low protein concentrations is unstable and tends to aggregate. This is prevented by the heterologous complex of p53 and HSP90 [18], probably involved in structural stabilization of the apo-form of p53 in the cytosol. In the mutated p53, a correction of the DNA binding pocket of p53 through small molecules binding may help overcome the negative consequences of p53 mutations. For instance, arsenic trioxide, which is used to treat acute promyelocytic leukemia, coordinates the three cysteine residues within DBD, distal to the zinc-binding site. The interaction stabilizes the loop-sheet-helix motif and the β-sandwich fold of the DNA-binding site, thus restoring transcriptional activity lost in the mutated p53 [19]. These findings demonstrate the importance of mechanistical knowledge of the structure–function relationship in p53 complexes, which may help in directing pharmacological regulation of the p53 function.

Apart from the central DNA-binding domain, a significant part of the p53 sequence is intrinsically disordered (Figure 1B). Proline-rich region (PRR) of p53 contains five copies of the tetrapeptide PXXP, providing for a highly flexible stretch of disordered structure, used in eukaryotic signaling modules. Deletion of this region impairs biological functions of p53, such as p53-dependent apoptosis [7]. Even if the regulatory complexes of proteins interacting with the p53 transactivation domains are formed, their regulation of p53 binding to DNA may be perturbed if no correct juxtaposition of p53 and DNA is achieved without the linker. 

Oligomerization, or tetramerization, region (TER, Figure 1A) participates in the p53 tetramerization required for DNA binding. Nuclear p53 is usually in a tetrameric state [20]. According to some studies, dimerization may occur under stress conditions when p53 interacts with Bcl-2 proteins on the outer mitochondrial membrane [21,22], although the non-nuclear p53 is supposed to be mostly monomeric.

The disordered N- and C-terminal regions participate in heterologous interactions of p53. Certain secondary structure elements of these disordered stretches are stabilized upon p53 binding to its protein partners (Figure 3). The process may involve so-called pre-structured motifs, i.e., unstable secondary structures transiently formed by the disordered regions generally represented by conformational ensembles [23]. The propensity of the disordered regions to form multiple conformations is demonstrated by different structures of the same region within different heterologous complexes of p53. For instance, this is obvious for the TAR1 region (in yellow) in the complex of p53 with P300 (Figure 3A, middle image) compared to the complex with the CREB-binding protein (CBP) (Figure 3A, image on the right). In the p53 complex with sirtuin 2, the C-terminal region of p53 forms a beta sheet (Figure 3B, image on the left), while in the complex with the Ca^2+^-dependent S100 protein, it forms an alpha-helix (Figure 3B, image on the right) [24].

Figure 3 shows that, depending on the p53 complex, Lys382 of the C-terminal region may be either acetylated (Figure 3B, image on the left) or not (Figure 3B, image on the right). The importance of multiple chemical modifications of p53 for a variety of structures acquired by its intrinsically disordered regions, is shown in Figure 4. The conformation of the C-terminal region in a complex with sirtuin 2 strongly depends on the acetylation of Lys382 (Figure 4, upper part) [25]. The biological relevance of structural destabilization of the non-acetylated peptide, whose acetylated state is bound to sirtuin 2, is obvious; the complex with the sirtuin 2 substrate, i.e., the Lys382-acetylated p53, has to be stabilized for deacetylation to occur. In contrast, the sirtuin 2 product, i.e., the deacetylated p53, should dissociate from sirtuin 2, which is supported by structural destabilization of the same peptide after Lys382 is deacetylated. In another complex of p53, i.e., the complex with p53-binding protein, the acetylation of neighboring Lys381 stabilizes the helical structure of the C-terminal domain of p53, which does not occur in the absence of acetylation (Figure 4, lower part) [26,27]. Thus, PTM may change the conformational ensemble and pre-structured motifs of the intrinsically unstructured regions stabilized in heterologous complexes. At the same time, the complex with p53-binding protein requires dimethylation of Lys382, while the complex with sirtuin 2 is formed with acetylated Lys382 (Figure 4). Thus, the competition between different types of PTM for the same residue may also regulate the formation of alternative p53 complexes and the corresponding functions. 

It is worth noting that the modifications of unstructured p53 regions are much more abundant compared to those of the folded DNA-binding domain (Figure 1A). This finding manifests an expectably higher exposure to PTM of the disordered vs. folded protein structures, as well as the PTM relevance for biological functions of p53. In fact, PTM in the disordered regions may support the formation of different heterologous complexes with p53, employing the same gene-encoded template. As a result, the intrinsically disordered regions with multiple PTM sites may form a much greater variety of specific heterologous complexes compared to the pre-existing binding pockets. Thus, the intrinsically unstructured regions of p53 and their PTM contribute to the complexity of p53 interactions with a great number of protein partners. 

Theoretically, the known propensity of p53 mutants to acquire new functions may employ a similar principle. Like PTM, mutations in the disordered regions may induce pre-structured motifs different to those of the wild-type p53 sequence. Conversely, structural destabilization due to mutations may confer the p53 sequence with other conformational ensembles. As a result, not only a perturbed interaction with one partner (loss of function), but also an acquired ability to bind another partner (gain of function) may be expected. Unfortunately, structural information regarding heterologous complexes with mutant p53 is mostly unavailable, often limited to in vitro pull-down studies using an expression of different parts of the p53 molecule. The latter approach is exemplified by a study of p53 DBD mutants that acquire the ability to bind p53 family members, p63 and p73, all sharing the DBD domain [28,29,30]. The formation of these nuclear heterologous complexes requires the mutated DBD of p53, as the heterologous complexes with p63 and p73 are not formed by the wild-type p53. However, the significance for binding of the regions beyond DBD has been quantified only for p63/p73, whose truncated isoforms demonstrate a weaker binding to the mutant p53 compared to full-length proteins [31,32,33]. Different effects on the binding to p63 and p73 are observed upon the D281G mutation of p53 [31,34]. As a result of the p63/p73 sequestration by mutated p53, the p63/p73 function as tumor suppressors is impaired, and oncogenicity is increased [35,36,37].

Mutations in the p53 DBD and the interactions of mutated p53 with nuclear proteins, such as transcription factors ETS2, NF-kB, HIF-1α, SMAD, SREBP or NF-Y, have been intensely studied [38] but are beyond the scope of this review, which is dedicated to non-nuclear complexes of wild-type p53. Nevertheless, mutated DBD of p53 may affect cellular proliferation, invasion, metastasis, stemness and metabolic reprograming, also through its interactions with non-nuclear proteins. For example, p53 DBD mutants R280K and R175H acquire the ability to bind DAB2IP in the cytoplasm [39].The ensuing interference with DAB2IP functions promotes proliferation in the prostate and breast cancer [40]. In H1299 cells, the interaction of p53 mutated at R175H, R248W or R273H with Rac1 prevents SENP1-dependent desumoylation and destabilization of Rac1, promoting tumor metastasis [41]. In contrast to wild-type p53 upregulating AMPK on the transcriptional level [42], p53 mutated at R175H binds the AMPKα subunit, thereby inhibiting the activation of AMPK by upstream kinases [43]. The R175H p53 interaction with AMPKα thus results in the activation of anabolic metabolism and the Warburg effect [44]. Thus, studies on the identification of structural determinants stabilizing different heterologous complexes of mutated p53 may reveal novel approaches to pharmacological regulation of the p53 functions gained through p53 mutations.

It is important to note that the affinities of p53 and its multiple isoforms, i.e., the products of the same gene, to the p53 targets greatly differ. Similar to the differences known for p53 interaction with different promoters [45], the different p53 affinities to the partner proteins contribute to hierarchical organization of the p53-activated programs, expressed as non-monotonous response to damage. For instance, the affinity of p53 to the promoters of cell cycle arrest and DNA repair genes is higher than to the promoters of apoptotic genes [45]. As a result, the latter are activated only if the activation of the former fails to correct the damage. As will be shown further, the different heterologous non-nuclear complexes of p53 similarly underlie the stepwise switches between various non-nuclear functions of p53 activated according to the damage level.

## 3. Distribution of Cellular p53 between Different Compartments

Cellular p53 is distributed between the nucleus, cytosol and mitochondria (Table 1), but the mitochondrial content is low in the absence of stress [46,47]. In the nucleus, p53 is found predominantly in its tetrameric form required for the transcriptional function, while in cytosol and mitochondrial matrix p53 is mostly monomeric [15,48].The different oligomerization of p53 in the compartments is not simply determined by protein concentration, as the difference is preserved even if the concentrations of p53 in these compartments are made equal [15]. Tetramerization promotes the nuclear accumulation of p53 by masking the nuclear export signal (NES), which remains exposed in the p53 dimer and monomer [49].

Under stress conditions, the dimers of p53 are accumulated at the outer membrane of mitochondria [21], which is essential for p53 interaction with Bcl-2 proteins [22]. In the mitochondrial matrix, p53 oligomerization into fibrillar structures may occur as a result of its interaction with prolyl peptidyl isomerase D (PPIF, cyclophilin D), known to catalyze p53 aggregation in vitro [50]. Although the exact mechanism of p53 transport into mitochondria is not known, multiple data point to the presence of p53 not only in the intermembrane space [16], but also in the mitochondrial matrix [46,47,51,52,53,54,55]. It is supposed that monoubiquitination of p53 is involved in its translocation to mitochondria [56]. These and other data suggest that p53 PTMs are strong determinants of its cellular compartmentalization [57,58]. To a great extent, this is due to the changes that PTMs induce in p53 protein interactions.

**Table 1 biomolecules-12-00327-t001:** Non-nuclear protein complexes of p53 and their role in p53 functions. The human protein identifiers, gene names and intracellular localizations are indicated, according to the Uniprot database.

Compartment	Interaction Partner	Unstressed Conditions	Stressed Conditions	p53 Region	Models and Methods Used	Additional Localizations of the Partner
Cytoplasm	NAD^+^-dependent protein deacetylase sirtuin 2 (Q8IXJ6, gene SIRT2) [25]	Destabilization of cytosolic p53 upon deacetylation of its K382. Regulation of protein–protein interactions of p53.		CTR	In vitro studies of p53 deacetylation by SIRT2, crystal structures of the bacterial homolog with the C-terminal p53 peptide comprising acetylated/deacetylated K382; p53 half-life assays	Nucleus
	Peptidyl-prolyl *cis-trans* isomerase NIMA-interacting 1 (Q13526, gene PIN1) [59]	n.d	Isomerization of p53 Pro47 to *cis* form, required for BAX activation	TAR	H1299 cell line expressing tamoxifen-inducible p53; NMR; large unilamellar vesicle permeabilization assay; in vitro studies of p53-PIN1 interaction and isomerization of p53 Pro47; apoptosis in cells with inhibited protein synthesis; inhibition of BCL2L1-dependent apoptosis to show the key role of PIN1 in activating BAX	Nucleus
	Ubiquitin carboxyl-terminal hydrolase 7 (Q93009, gene USP7) [56,60,61,62]	Deubiquitinates and stabilizes p53	Camphotericin-induced monoubiquitination of p53 leads to its mitochondrial translocation and subsequent deubiquitination by USP7, which may lead to transcription-independent apoptosis	CTR	HCT116, RKO and ML cells; manipulated expression of USP7 (+/+ and -/- cells); p53 increases in the USP7 knockout cells in vitro p53 deubiquitination assay; crystal structure of USP7-p53; subcellular fractionation and IP; camptothecin treatment	Nucleus, mitochondria
	E3 ubiquitin-protein ligase parkin (O60260, gene PRKN) [63]	n.d	P53 binds PRKN and retains it in cytosol, preventing PRKN-dependent autophagy signal transduction	DBD	MEFs, Hl-1 cells, rat neonatal cardiomyocytes, mouse heart lysates; IP; p53 overexpression in MEFs; nutlin and doxorubicin treatments in Hl-1	Nucleus, endoplasmic reticulum, mitochondrion
	Glucose-6-phosphate 1 dehydrogenase (P11413, gene G6PD) [15]	G6PD binds almost all cytosolic p53. P53 inhibits G6PD via lasting physical and transient catalytic interactions	Fraction of overexpressed p53 is free of G6PD. G6PD is inhibited by TIGAR, which is a transcriptional target of p53	CTR	Mouse embryonic fibroblasts (MEFs), mice tissues and cancer cell lines; p53-/- MEFs and cancer cells; pull-down assays in cancer cells; the p53 effect on G6PD dimerization in MEFs and cancer lines; ratios of G6PD and p53 in HCT116 p53+/+ cells are determined to depend on doxorubicin treatment (100:3 and 10:1 in the unstressed and doxorubicin-treated cells, correspondingly)	Cellular membrane
	Heat shock protein 90 kDa alpha (P08238, gene HSP90AB1) [18,64,65]	Stabilizes the Zn^2+^-free p53, folds p53. Inhibition of HSP90 promotes p53/PUMA/BAX-mediated apoptosis in p53 wild type cells		DBD	In vitro experiments: NMR spectroscopy and gel electrophoresis show no significant difference between zinc-free and holo-p53 with high concentrations of p53; aggregation assay of zinc-free p53 at physiological p53 concentrations shows decrease in aggregation of unstable zinc-free p53 by HSP90 addition. Colorectal cancer cell lines, also with manipulated expression of p53, PUMA and apoptotic proteins	Nucleus, Hsp90 paralogs in ER (HspC4) and mitochondria (HspC5)
	Heat shock 70 kDa protein 1 (P0DMV8, gene HSPA1) [65]	Destabilizes p53 by unfolding		DBD	In vitro experiments	
	Ribonucleoside-diphosphate reductase subunit M2 B (Q7LG56, gene RRM2B) [66]	P53 binds RRM2B and retains it in cytosol	RRM2B is liberated from p53 and translocated to the nucleus, where it promotes DNA reparation	n.d.	KB cell line; immunoprecipitation (IP) and colocalization with or without UV irradiation	Nucleus
	Thioredoxin (P10599, gene TXN) [67,68]	Reduced thioredoxin enhances p53 transcriptional activity, while oxidized thioredoxin inhibits p53 transcriptional activity		n.d.	WiDr, MG63, HeLa cells; WiDr and MG63 cells with transient expression of TXN; in vitro electrophoretic mobility shift assay of p53–DNA binding; fluorescent microscopy. Yeast; thioredoxin and/or thioredoxin reductase mutations; in vitro binding assay and one-hybrid assay; p53 cysteines substitutions KD hTXN = 0.9 uM	
	Focal adhesion kinase 1 (Q05397, gene PTK2) [69]	Inhibition of cytosolic p53 by PTK2 promotes survival. A feedback loop mechanism of regulation of p53 and PTK2		PRR	Cancer cells; IP, pull-down and confocal microscopy methods	Nucleus, extracellular space
	14-3-3θ (P27348, gene YWHAQ) [70]	Low content of the complex	Highly abundant complex	C277	Diamide treatment; mass spectrometry	
	14-3-3σ (P31947, gene SFN) [71]	n.d.	Upon adriamycin and ionizing radiation treatment, binding of p53 to 14-3-3σ increases p53 half-life	CTR	A549, R1B, L17, 293T cells; IP; adriamycin and radiation treatments; immunofluorescence; p53 half-life assays; overexpression of 14-3-3σ; pulse-labeling with [^35^S]methionine	Nucleus
	Clathrin heavy chain 1 (Q00610, gene CLTC), Epidermal growth factor receptor(P00533, gene EGFR) [72]	n.d.	Interacting with CLTC and/or EGRF, p53 promotes epidermal growth factor (EGF) internalization through clathrin-mediated endocytosis	n.d.	H1299 and TIG-7 cells; transfection with p53 construct; p53 knockdown; IP; p53-CLTC colocalization; EGF internalization assay	Cellular membrane and vesicles
Endoplasmic reticulum (ER)	Sarcoplasmic/endoplasmic reticulum calcium ATPase 2 (P16615, gene ATP2A2) [73]	n.d.	p53 enhances ATP2A2 activity, causing mitochondrial Ca^2+^ overload	CTR	MEFs and H1299 cells; ER vesicles isolation; p53 -/- and p53+/+ mice; pull-down assay; coprecipitation assay; Ca^2+^ accumulation kinetics in ER vesicles; doxorubicin treatment	
Mitochondrial outer membrane	Apoptosis regulator BAX (Q07812, gene BAX) [74]	No p53 complex formation to activate BCL2L1	Activation of BCL2L1 in a complex with p53 promotes mitochondrial outer membrane permeabilization (MOMP) and apoptosis	PRR, DBD	MEFs cells; in vitro experiments with mutant proteins; NMR; knockout and overexpression of BAX and/or p53 and its various mutant forms	Cytoplasm
	Bcl-2-like protein 1 (Q07817, gene BCL2L1) [22]			DBD	Crystallization of dimeric p53 DBD with BCL2L1	Mitochondrial inner membrane, matrix, cytosol, cytosolic side of nuclear membrane
	Mitochondrial E3 ubiquitin ligase 1 (Q969V5, gene MUL1) [75]	Ubiquitinates p53 at Lys24 for proteasomal degradation, thus negatively affecting both nuclear and cytoplasmic functions of p53		TAR2	H1299, MCF10A, NHLF, U2OS, MCF cells; manipulated expression of MUL1; proteasome inhibition; NMR structural studies of the RING domain of MUL1 and TAR2 of p53; in vitro experiments; pull-down assay and IP; ubiquitination assay in vitro and in vivo	
Mitochondrial intermembrane space (IMS)	Mitochondrial intermembrane space import and assembly protein 40 (Q8N4Q1, gene CHCHD4) [16]	Colocalization of p53 with CHCHD4 increases and decreases according to the manipulated expression of CHCHD4	Overexpressed CHCHD4 decreases nuclear p53, increasing mitochondrial colocalization of p53 and CHCHD4	n.d.	HCT116 cells and primary human myoblasts; manipulated expression of CHCHD4 and p53, treatment of cells with H_2_O_2_	
Mitochondrial inner membrane	ATP synthase subunit O, mitochondrial (P48047, gene ATP5PO) [47]	ATP5PO is activated by p53 mitochondrial localization, induces F1-F0 ATP-synthase assembly		n.d.	HCT116 and H1299 cells; IMS and matrix p53 localization; IP and LC-MS identification of ATP5PO-p53 interaction; etoposide treatment; F1-F0 quantification	
Mitochondrial matrix	Lon protease homolog, mitochondrial (P36776, gene LONP1) [51]	LONP1 interaction with p53 is observed under normal conditions and during oxidative stress, likely regulating the availability of p53 to interact with its other targets both inside and outside mitochondria		DBD	HSC3 and 293T cells; p53-LONP1 colocalization in H_2_O_2_-treated cells; pull-down assay; IP; rotenone treatment, LONP1 and/or p53 overexpression and knockdown	
	Single-stranded DNA-binding protein, mitochondrial (Q04837, gene SSBP1) [52]	n.d.	SSBP1 enhances 3′-exonuclease activity of p53, promoting base excision repair (BER), presumably activated under stress	TAR1,2	In vitro experiments using purified proteins	Nucleoid
	Transcription factor A, mitochondrial (Q00059, gene TFAM) [46]	n.d.	TFAM is guided by p53 to damaged regions of mtDNA	CTR	KB and HCT116 cells; IP; cisplatin and 5-fluorouracyl treatments; pull-down assay	Nucleoid
	DNA polymerase subunit gamma-1 (P54098, gene POLG) [53]	POLG is activated by p53 independent of ethidium bromide treatment		n.d.	ML-1 and HCT116 cells; IP and colocalization; ethidium bromide treatment, in vitro assays of p53 influence on POLG activity	Nucleoid
	Peptidyl-prolyl *cis-trans* isomerase F, mitochondrial (P30405, gene PPIF) [54]	Free from p53, PPIF does not induce formation of permeability transition pore (PTP)	PPIF interacts with p53, inducing formation of PTP and necrosis	DBD	MEFs and HCT116 cells; p53 or PPIF -/- MEFs; p53 or PPIF+/- mice; IP; H_2_O_2_ treatment; induction of necrosis by targeting p53 to mitochondrial matrix	
	Superoxide dismutase [Mn], mitochondrial (P04179, gene SOD2) [55]	No complex detected	SOD2 inhibition by p53 causes overproduction of mitochondrial ROS	n.d	JB6 cells and mouse skin epidermis; IP; 12-O-tetradecanoylphorbol-13-acetate treatment; SOD2 assay	

## 4. Heterologous Complexes of p53 with the Enzymes Regulating Post-Translational Modifications of p53

As noted above, p53 has multiple sites for PTMs that regulate the p53 interactions with partner proteins and, hence, the biological functions of p53. A local preference for certain modification type may be noticed (Figure 1A); p53 phosphorylation is most abundant in the N-terminal transactivation region 1 (TAR1). In contrast, acetylation of p53 mainly occurs at the C-terminal region (CTR). Phosphorylation brings about a negative charge of the phosphate residue, while acetylation attenuates the positive charge of a lysine residue. Several proteins involved in these PTMs have multiple intracellular locations, reacting with p53 not only in the nucleus, but also in cytosol. In particular, they include the NAD^+^-dependent deacetylase sirtuin 2, catalyzing a removal of acetyl residues from p53 lysine residues, and focal adhesion kinase 1 (PTK2), phosphorylating p53 [69].

Protein interactions may be strongly affected due to electrostatic changes in PTMs. For instance, the phosphorylation of the N-terminal transactivation domain of p53 promotes a dissociation of the complex between p53 and its negative regulator Mdm2; occurring in response to DNA damage, this PTM results in p53 stabilization and activation [76,77,78,79]. In the p53 C-terminus, multiple lysine residues of p53 become acetylated in response to DNA damage [80,81,82]. The synergistic or antagonistic effects of different modifications of p53, promoted by different cellular conditions, are involved in the complex outcome of the p53-dependent transcriptional and non-transcriptional programs. In this regard, the role of ubiquitination in the stability and trafficking of p53 deserves special attention. Ubiquitination of p53 is catalyzed by ubiquitin ligases of a different location. While p53 polyubiquitination at C-terminal lysine residues destabilizes p53 by targeting the protein for proteasomal degradation, monoubiquitination of p53 has an activating role in the transcription-independent mitochondrial function, such as the p53-dependent death program [83]. The ubiquitination of p53, catalyzed by the RING finger domain of mitochondrial ubiquitin ligase MUL1 at the outer mitochondrial membrane, is probably involved in the regulation of p53 transport into mitochondria [75,84]. Ubiquitin-specific protease (USP7) removes the ubiquitin residue(s), strongly stabilizing cytosolic p53 [62].

The formation of intramolecular disulfides is an important PTM of p53, coupling its function to the cellular redox state. DNA binding of p53 increases under reducing conditions and is abrogated upon cysteine residues alkylation [68,85,86,87,88]. Cytosolic thioredoxin (TNX) may reduce p53 disulfides in vitro and in vivo, thus regulating p53 transcriptional activity [67,68]. TNX overexpression is known to enhance p53-dependent reporter gene expression [68]. However, transactivation is inhibited in budding yeast lacking the *TRR1* gene encoding thioredoxin reductase. It thus appears that the accumulation of oxidized thioredoxin inhibits p53 function [67]. This assumption is supported by suppression of the inhibitory effect of thioredoxin reductase deletion through the accompanying deletion of TNX [67].

Substitutions of Zn^2+^-coordinating Cys176, Cys238, Cys242 or Cys275 for serine residues inactivates p53 [67]. NMR studies of p53 after H_2_O_2_ treatment in vitro reveal the formation of intramolecular disulfide bonds between Cys182 and Cys238, as well as Cys176 and Cys242 [89]. The findings suggest that the cellular redox state regulates p53 activity through the TNX-dependent reduction/oxidation of the forementioned cysteine residues.

Peptidyl prolyl *cis-trans* isomerases modify protein conformation by catalyzing the isomerization of peptide bonds involving proline residues. This single reaction causes significant structural changes by altering the relative positions of protein parts. The isomerase PIN1 localized to cytosol mediates p53-dependent BAX activation [59]. In the mitochondrial matrix, another peptidyl prolyl isomerase, PPIF, promotes mitochondrial permeability transition after binding to p53 [54].

## 5. Other Interaction Partners of p53 Outside the Nucleus

Most of the p53 non-nuclear complexes mentioned in Table 1 are schematically depicted in Figure 5. Figure 5A refers to standard conditions of low p53 expression, while Figure 5B shows the same and additional interactors under cellular stress when p53 is upregulated. In the former case (Figure 5A), only the p53 complexes with abundant and/or high affinity partners are pronounced. Due to the stress-associated upregulation of p53, many more of its non-nuclear interactions are enabled (Figure 5B) compared to the standard conditions (Figure 5A). Some of the complexes in Table 1, merely those studied in vitro or whose role in the stress response is not sufficiently different or clarified, are omitted from Figure 5 for clarity.

### 5.1. Cytosol

Heat shock proteins 90 and 70 kDa participate in dynamic folding and unfolding of p53, respectively, thus controlling the availability of functional p53 for interaction with nuclear and non-nuclear targets [18,64,65].

Under normal conditions (Figure 5A), almost all folded cytosolic p53 is bound to glucose-6′-phosphate dehydrogenase 1 (G6PD), in view of the G6PD excess over p53 [15]. G6PD is an enzyme of the oxidative part of pentose phosphate pathway providing phosphoribose and reducing equivalents in the form of NADPH for biosynthesis. On the transcriptional level, the pathway is regulated by p53 through the p53-dependent induction of TIGAR protein, which directs the carbohydrate degradation from glycolysis to the pentose phosphate pathway under a low level of DNA damage [90]. The formation of the p53 complex with G6PD represents an additional control point in the p53-dependent regulation of pentose phosphate pathway. The two proteins in the complex demonstrate mutual negative regulation [15]. G6PD keeps p53 unavailable for nuclear translocation and/or interaction with other targets, whereas p53 prevents the formation of G6PD dimer required for the enzyme activity. Moreover, p53 binding also exerts a transient inhibition of G6PD, i.e., adverse effects of the complex with p53 on G6PD dimerization are transiently preserved after the complex dissociation [15]. This is probably due to the dimerization-incompetent conformation and/or PTM of G6PD in its complex with p53. Owing to this transient effect, p53 inhibits G6PD significantly more than is expected from the molar ratio of p53 to G6PD [15]. Only when p53 expression is increased (Figure 5B) does p53 also become available for the formation of other heterologous complexes. Under these conditions, the flux through G6PD also increases due to the TIGAR-dependent activation of the pathway. 

Stress-induced upregulation of p53 causes its binding to E3 ubiquitin-protein ligase parkin in the cytoplasm (Figure 5B), which blocks the parkin translocation into mitochondria required for the parkin-induced mitophagy [63]. This non-nuclear function of p53 is also linked to the nuclear one, where p53, on the contrary, activates the transcription of the autophagy genes (ref [91]) [92]. The p53-parkin complex is controlled by acetylation. In the heart of the SIRT3^−/−^ mice lacking the mitochondrial deacetylase, acetylation of p53 at K317 is increased. Compared to the wild-type mice, the increased acetylation in SIRT3-/- mice stabilizes the p53-parkin complex and hampers parkin translocation into mitochondria in response to CCCP treatment [93].

Clathrin-mediated EGF endocytosis is accelerated by cytosolic p53, whose translocation to cellular membrane is elevated 4-fold upon the addition of EGF to the cells; on the contrary, in the absence of p53, EGF internalization is significantly reduced [72]. In view of the involvement of EGF in cellular stress response [94], which is also associated with p53 upregulation, this function of p53 may synergistically support EGF participation in stress response. 

Cytosolic complex of p53 with focal adhesion kinase PTK1 is known in addition to the more characterized nuclear complex. The cytosolic complex is supposed to participate in the feedback loop regulation mechanism of p53 action in cell survival [69].

### 5.2. Endoplasmic Reticulum

Under stress conditions, p53 activates sarcoplasmic/endoplasmic reticulum calcium ATPase 3 (SERCA) [73]. This may help to overcome the endoplasmic reticulum (ER) stress known to occur upon SERCA inhibition [95]. ER stress is a condition where ER’s capacity to fold proteins is compromised, followed by unfolded protein response (UPR) [96]. If ER stress continues, the cells may commit apoptosis via several pathways, including the BCL-2-dependent mitochondrial pathway [95]. Under these conditions, p53-dependent SERCA activation may accelerate the mitochondrial calcium overload and apoptosis [97,98,99,100] (Figure 5B), in view of the existence of mitochondrial contact sites with ER [101] (Figure 5).

Thus, p53 participates in ER/UPR-induced stress response, contributing to cell death by both transcriptional activation of proapoptotic genes and SERCA activation. At the low levels of ER stress, the activation may help in overcoming ER stress, but if the stress persists, the p53-dependent SERCA activation may cause mitochondrial Ca^2+^ overload, promoting apoptosis.

### 5.3. Mitochondria

Under stress, the cytosolic p53 substitutes, BAX and BID, in their complex with BCL-xl on the outer mitochondrial membrane, thus releasing these proapoptotic proteins for initiating apoptosis [22]. In addition, p53 interaction with BAX changes the BAX conformation, resulting in BAX homo-oligomerization promoting mitochondrial outer membrane permeabilization (MOMP) [74]. According to a recent crystallographic study, the dimer of p53 DNA-binding domain interacts with BCL-xl [22], with a mutation on the dimer interface blocking the interaction and apoptosis. However, the complex of monomeric BCL-xl and p53 DNA-binding domain is also crystallized [102]. In both cases, the proteins are manipulated to obtain the resolved crystals. In the former study, BCL-xl lacks the disordered loops and C-terminal region; in the latter—only the C-terminal region, but the p53 DNA-binding domain carries triple mutations. Obviously, all these artificial structural changes may affect the formation of the complex and its stoichiometry.

The activation of BAX requires *cis*-configuration of the peptide bond at Pro47 in the proline-rich region of p53 [59]. As mentioned above, the isomerization from a more stable *trans*-configuration is catalyzed by cytosolic PIN1 [59]. Thus, apoptosis is controlled via the conformational change of the cytosolic p53.

A component of mitochondrial oxidative transport system, the disulfide-comprising mitochondrial import and assembly protein 40 (CHCHD4), translocates p53 to mitochondria with the intermediary formation of an intermolecular disulfide between CHCHD4 and p53 in the intermembrane space (Figure 5B) [16]. The pathway implies the intramitochondrial release of p53 in the disulfide form, while CHCHD4 is reoxidized by cytochrome c oxidase [103]. A manipulated expression of cellular CHCHD4 or impairment of cytochrome c oxidase affect intracellular distribution of p53 between the nucleus and mitochondria [16]. In vivo, physical activity causes a FOXO3-mediated decrease in CHCHD4 expression, which increases nuclear p53 at the expense of mitochondrial one [104], reciprocating the in vitro events upon the increased CHCHD4 expression [16].

Although the mechanisms of p53 transport into the mitochondrial matrix are not fully clarified, several matrix proteins are direct binding partners of p53. The partners are involved in the maintenance of mitochondrial genome stability [46,52,53], energy metabolism [47] and cell death [54,55].

In the mitochondrial matrix, p53 is sequestered by mitochondrial LON protease homolog (LONP1) (Figure 5). Under oxidative stress conditions, this decreases nuclear p53 content and stabilizes mitochondrial p53, promoting cell survival [51]. Such interaction may also be involved in the regulation of p53 binding to other matrix partners, such as necrosis-inducing peptidyl-prolyl *cis-trans* isomerase (PPIF) [54].

The interaction of p53 with single-stranded DNA-binding protein (mtSSB) enhances 3’-exonuclease activity of p53, presumably promoting the base excision repair (BER) in mitochondria [52] (Figure 5B). Colocalization of p53 and mitochondrial DNA polymerase gamma increases the latter’s activity [53]. In addition, p53 guides the binding of mitochondrial transcription factor A (TFAM) to damaged DNA [46]. The known positive correlation between p53 translocation to mitochondrial matrix and the level of mitochondrial DNA damage [105,106] may support these interactions of p53 involved in the protection of mtDNA.

Mn^2+^-dependent mitochondrial superoxide dismutase (SOD2) is inhibited by p53 in the mitochondrial matrix (Figure 5B). As a result, reactive oxygen species increase, activating nuclear p53 and p53-dependent apoptosis [55]. The dual role of reactive oxygen species in signaling the perturbed homeostasis for its correction or inducing cell death may thus underlie a similar dualism of the p53-dependent outcomes.

The peripheral stalk subunit O of ATP synthase (ATP5PO, oligomycin sensitivity conferral protein OSCP) is activated by p53 [47], presumably promoting mitochondrial function under stress conditions when p53 is accumulated in the mitochondria (Figure 5B). 

Peptidyl-prolyl *cis-trans* isomerase F (PPIF, Cyclophilin D, CypD), an inductor of permeability transition pore (PTP), interacts with p53 in the mitochondrial matrix (Figure 5B). The interaction induces necrosis under oxidative stress [54]. According to a popular model of PTP, ATP synthase is a major channel mediating the permeability transition [107], with PTP occurring when the ATP5PO subunit interacts with PPIF [108]. As discussed above, p53 interacts with PPIF and regulates ATP5PO [47]. Hence, competitive binding of p53 to PPIF and ATP5PO may regulate ATP synthase and its switching to the PTP mode. 

### 5.4. Heterologous Complexes Coupling the Redox-Dependent Modifications of p53 Cysteine Residues to the Action of Ribonucleotide Reductase in Different Cellular Compartments

As discussed above, p53 may be regulated by reversible oxidation of its cysteine residues. This feature confers the redox sensitivity to the nuclear, cytosolic and mitochondrial complexes of p53. The DNA-binding surface of p53 includes many of the DBD cysteine residues (Figure 2). Three of the cysteine residues in the p53 monomer coordinate Zn^2+^ required for the p53 interaction with DNA. A loss of Zn^2+^ destabilizes the p53 structure and increases the probability of cysteine residues oxidation into intramolecular disulfides. Under oxidative stress modeled by the diamide treatment, an intermolecular disulfide between p53, C277 and 14-3-3θ has been identified [70]. However, the physiological relevance of this finding is unclear, as the p53 interactions with 14-3-3 proteins are mostly known to increase DNA binding of p53 [109]. The increased binding is not possible if the p53 disulfide is formed with C277, the cysteine residue neighboring DNA (Figure 2).

As discussed above, thioredoxin reduces p53 oxidized at its cysteine residues, regenerating the latter from their disulfides. On the other hand, reduced thioredoxin is a substrate of ribonucleotide reductase, providing deoxyribonucleotides for DNA synthesis. In the cytoplasm of non-damaged cells, one of the ribonucleotide reductase subunits, ribonucleoside-diphosphate reductase subunit M2 B (RRM2B, p53R2), is sequestered by p53 from the formation of a catalytically active enzyme. Under UV stress, RRM2B is released from its complex with p53 and translocates to the nucleus, where it is suggested to form an active ribonucleotide reductase enzyme with the other subunit (Figure 5 and Figure 6). As a result, DNA reparation may be supported by the on-site synthesis of deoxyribonucleotides [66]. Nuclear p53 also activates the transcription of the RRM2B gene [110]. Thus, nuclear and cytosolic p53s organize the DNA stress response in accord (Figure 6). Although the nuclear translocation of RRM2B for nuclear synthesis of deoxyribonucleotides has been observed in different studies [111,112,113,114], these findings are still debated. Some research groups insist on the solely cytosolic action of ribonucleotide reductase [115] and consider the nuclear translocation of RRM2B an experimental artifact. Indeed, the translocation of one subunit of the enzyme is not sufficient for synthesis to occur, as it also requires the other subunit, thioredoxin and thioredoxin reductase (Figure 6). Nevertheless, the model of DNA reparation with the on-site reduction of ribonucleotides, shown in Figure 6, is implied by independent data [116]. We hypothesize that the redox sensitivity of p53 cysteine residues and their reduction by thioredoxin may be employed for the regulation of the whole process, as presented in Figure 6. This hypothesis is supported by the physical interaction of thioredoxin with p53 in vitro and in vivo, and a decreased transactivation by p53 in cells with disabled thioredoxin reductase [67].

Mutations in RRM2B gene induce mtDNA depletion [119]. On the other hand, RRM2B is known to interact with mitochondrial thioredoxin reductase (TXNRD2), causing its activation, which contributes to cell survival and proper functioning of mitochondria [117]. This activating interaction may be perturbed upon mutations of RRM2B, increasing deleterious effects of the mutation on synthesis of mitochondrial DNA. As shown in Figure 6, it may be suggested that thioredoxin couples the redox-dependent regulation of p53 and deoxyribonucleotide (dNTP) synthesis in not only nuclear, but also mitochondrial compartments. In the mitochondrial matrix, RRM2B-dependent activation of TXNRD2 [117] increases the level of mitochondrial thioredoxin (TNX2) reduction, stimulating both the mitochondrial dNTPs production for the mitochondrial DNA repair and TNX2-dependent reduction of intramitochondrial p53 to its active form. It is also shown that TNX2 prevents PPIF from the induction of apoptosis by formation of a redox-dependent complex, potentially competitive to the formation of the heterologous complex of p53 and PPIF [118]. While the p53-PPIF complex promotes apoptosis [54], the TNX2-PPIF complex is associated with survival [118]. Finally, in the mitochondrial intermembrane space, p53 forms a mixed disulfide with CHCHD4, whose role in p53 signaling is unclear, but the accumulation of p53 colocalized with CHCHD4 decreases the nuclear content of p53 [16]. As a component of the disulfide relay system, CHCHD4 participates in the oxidative folding of mitochondrial proteins. The heterologous complex between p53 and CHCHD4 [16] implies that CHCHD4 is involved in p53 transport into the mitochondrial matrix, in which case p53 will appear in the matrix in its oxidized form. As a result, the oxidized intramitochondrial p53 would require reduction through mitochondrial thioredoxin to initiate the redox-dependent functions, such as mitochondrial DNA repair.

Thus, several known heterologous complexes of not only p53, but also its partner proteins, suggest that the redox regulation of p53 by thioredoxin may be similarly coupled to the ribonucleotide reductase function in both the nuclear and mitochondrial compartments (Figure 6). Moreover, the failure in DNA repair under persistent oxidative stress may be translated into death programs involving permeability transition through the competitive redox-sensitive interaction of PPIF with p53 or TNX2 (Figure 5) or of p53 with PPIF or ATP5PO (see Section 5.3).

## 6. Heterologous Complexes in Hierarchical Organization of the p53-Inducible Responses

Our analysis of the published data on the formation of non-nuclear p53 complexes exposes the role of these complexes in such an important feature of p53-dependent regulation as a hierarchical organization of the response according to the damage level. In contrast to the monotonous response based on consistently accumulated damage, the hierarchy of p53 interactions, exemplified by a threshold-dependent representation of its many non-nuclear complexes (Figure 5), discriminates between the reparable and irreparable states without a long accumulation of a damage, but through cooperative multipoint control and information transfer between different cellular compartments. In particular, our analysis exposes that many of the studied non-nuclear complexes of p53 are involved in the pathways that are targets of the p53 transcriptional regulation. For instance, cellular p53 responses are regulated by cytosolic and nuclear PTK2 (Table 1) or through transcriptional (activation of the genes) and non-transcriptional (interaction with parkin, Figure 5) control of p53-dependent autophagy. As a result, non-nuclear heterologous complexes of p53 support regulatory loops for additional control of the p53-organized response according to specific cellular conditions. The advantages of this sensitive network-supported discrimination between the repair and death programs include not only sparing the resources, but also preventing the damage spreading from compromised cell(s) to the neighborhood. Failure to initiate this discrimination timely is especially dangerous for multicellular organisms, as they may develop problems at the systemic level. Such problems are exemplified by metastases, septic shock, etc. Hence, increased sensitivity to metabolic disbalance is usually inherent in eukaryotes vs. prokaryotes. In general, such an increase is achieved by introducing additional components into the regulatory network. In particular, increased sensitivity is observed when activation and deactivation of a sensor is performed by separate components instead of a single autoregulated process [120]. In this regard, the p53 “holders”, such as G6PD and HSP proteins in the cytosol or LONP1 in mitochondria, represent components of the p53 activation, additional to its expression. HSP90 and HSP70 control the level of folded p53 by their dynamic stabilization and destabilization, respectively [65]. Binding almost all of the folded p53 in cytosol and mitochondria until a threshold in folded p53 level is achieved, G6PD and LONP1 are the next-level guardians of the p53 transcriptional and non-nuclear functions; only after the threshold is overcome are p53 interactions with other targets possible (Figure 5). In particular, the heterologous cytosolic complex between p53 and G6PD [15] represents a regulatory loop to perform hierarchical adjustment of the TIGAR expression, pentose phosphate pathway activation and other p53-dependent programs to the damage level (Figure 5). Under normal conditions, all cytosolic p53 is bound to G6PD. Under damaging conditions, increasing levels of expressed p53 may first activate the pentose phosphate pathway through TIGAR, whose p53-dependence leads to feeding the pentose phosphate pathway at the expense of glycolysis. Increasing damage will further increase the p53 available for other protein interactions (Figure 5B), initiating other programs, such as p53-dependent apoptosis. A similar p53 “holder” function may be performed by LONP1 within mitochondria (Figure 5). 

## 7. Conclusions

A multitude of p53 functions, including nuclear and non-nuclear ones, are strongly dependent on heterologous complexes of p53, formed outside the nucleus. Mutual activation or inhibition of p53 and its protein partners may occur in their heterologous complexes, which are also significant for sequestration of component proteins from other interactions or in different compartments. The non-nuclear heterologous complexes of p53 participate in hierarchical regulation of the p53-induced responses through (i) controlling the p53 post-translational modifications; (ii) transferring the information on the cellular state through p53 trafficking between different cellular compartments; (iii) activating different p53-dependent programs according to the level of p53 and/or its partners’ expression and their affinities to each other. 

## Figures and Tables

**Figure 1 biomolecules-12-00327-f001:**
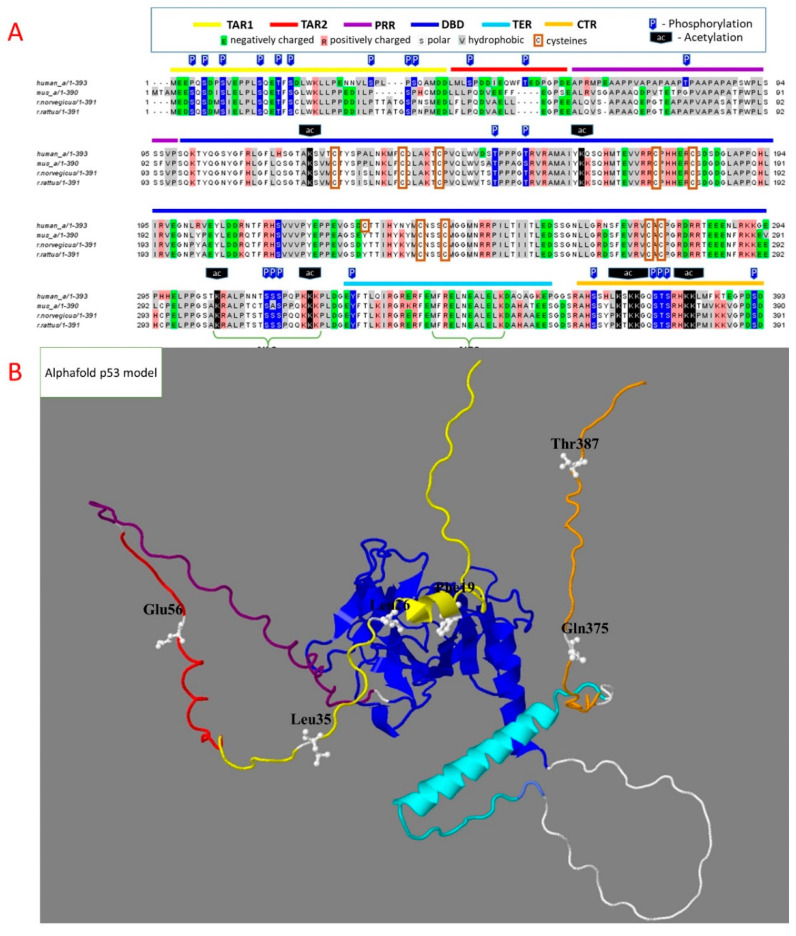
Structural determinants of p53 homo- and heterologous protein complexes. Throughout this and other figures, specific regions are highlighted by color: yellow—transactivation region 1 (TAR1), red—transactivation region 2 (TAR2), purple—proline-rich region (PRR), blue—DNA-binding domain (DBD), cyan—tetramerization region (TER), orange—C-terminal region (CTR). (**A**)—Sequence of human p53 is aligned to that of model organisms, mouse and rat. Acetylation and phosphorylation sites with more than five references, according to the Phosphosite database, are marked as P (blue arrows) and Ac (black arrows) above the sequences. The cysteine residues are framed. (**B**)—The 3D model of full-length p53 structure, predicted by AlphaFold (https://alphafold.ebi.ac.uk/entry/P04637, accessed on 6 January 2022). The marked residues define the regions acquiring secondary structure in heterologous protein complexes.

**Figure 2 biomolecules-12-00327-f002:**
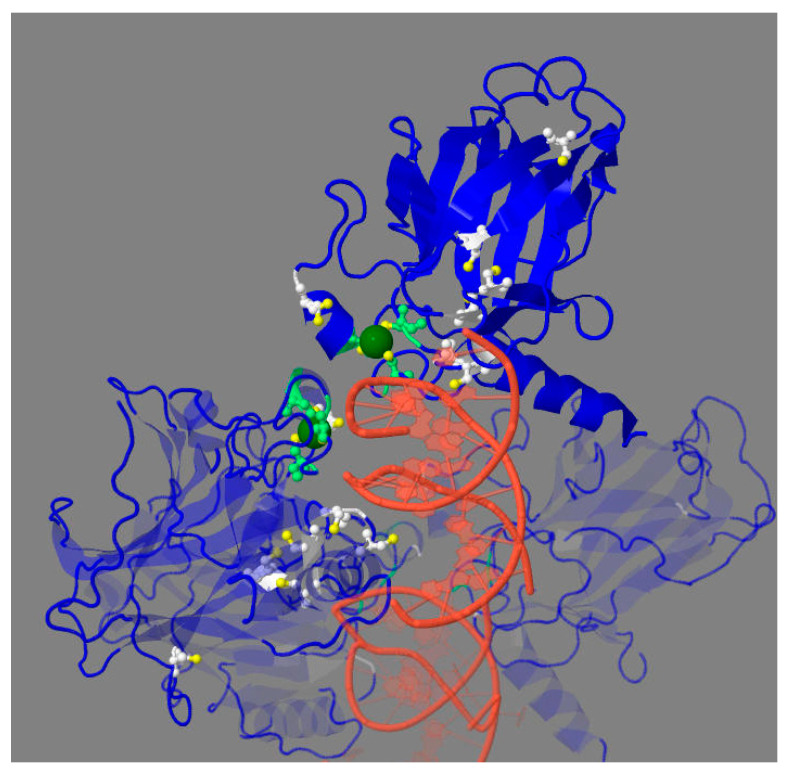
DNA binding of tetrameric DBD of p53 (PDB: 2AC0). Zn^2+^ ion—in green, sulfur atoms—in yellow, the Zn^2+^ -coordinating cysteine residues—in light green, other cysteine residues of DBD—in white. Except for the less conserved Cys229, all cysteines are located on the DNA-binding surface of DBD. For clarity, the cysteine residues and Zn^2+^ ions are shown only in two out of four DBDs.

**Figure 3 biomolecules-12-00327-f003:**
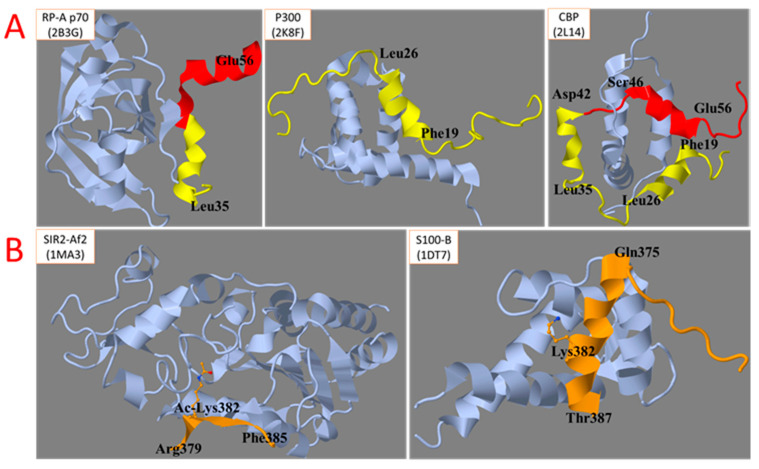
Structural changes of disordered regions of p53 upon the formation of heterologous protein complexes. (**A**)—N-terminal region. (**B**)—C-terminal region. The Uniprot names of the p53 partner proteins and pdb identifiers of the structures of their complexes with p53 regions involved in the interactions are indicated in the upper-left corner of each image.

**Figure 4 biomolecules-12-00327-f004:**
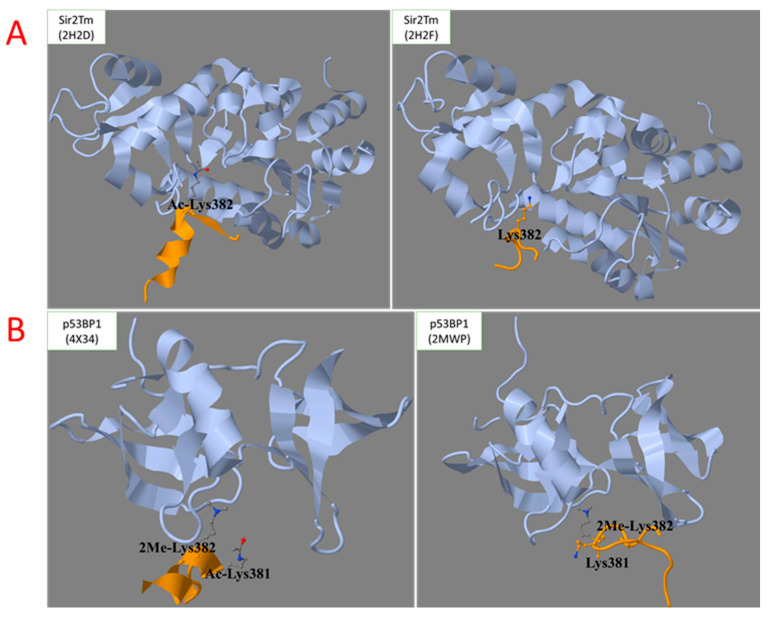
The role of post-translational modifications of p53 in the stabilization of secondary structures of the intrinsically disordered C-terminal region upon the heterologous complex formation with NAD^+^-dependent protein deacetylase Sir2Tm (**A**) and TP53-binding protein 1, p53BP1 (**B**). The Uniprot names of the p53 partner proteins (shown in grey) and pdb identifiers of the structures of their complexes with p53 C-terminal region (in yellow) are indicated in the upper-left corner of each panel.

**Figure 5 biomolecules-12-00327-f005:**
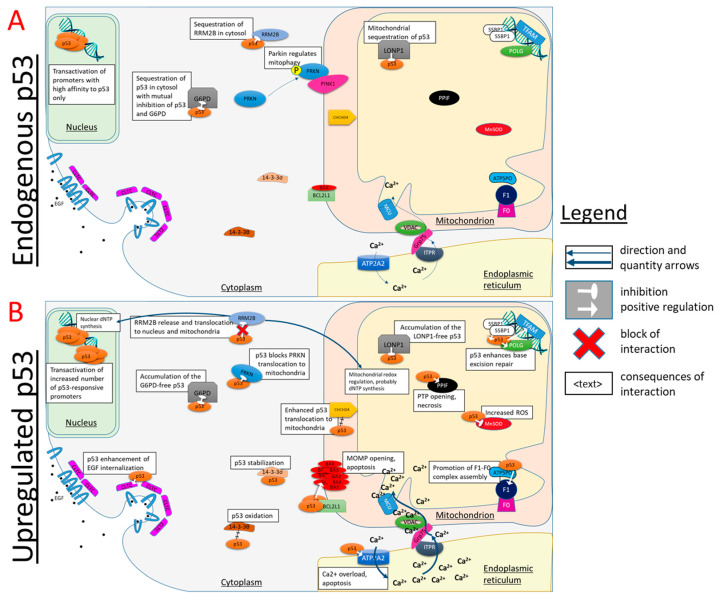
Involvement of non-nuclear protein complexes of p53 in p53-dependent regulation of cellular homeostasis. The complexes formed upon endogenous p53 expression, (**A**) or in case of p53 upregulation during cellular stresses, such as oxidative, cytotoxic or genotoxic ones (**B**), are shown. The names of cellular compartments are underlined. Further explanations are given in the text and Table 1.

**Figure 6 biomolecules-12-00327-f006:**
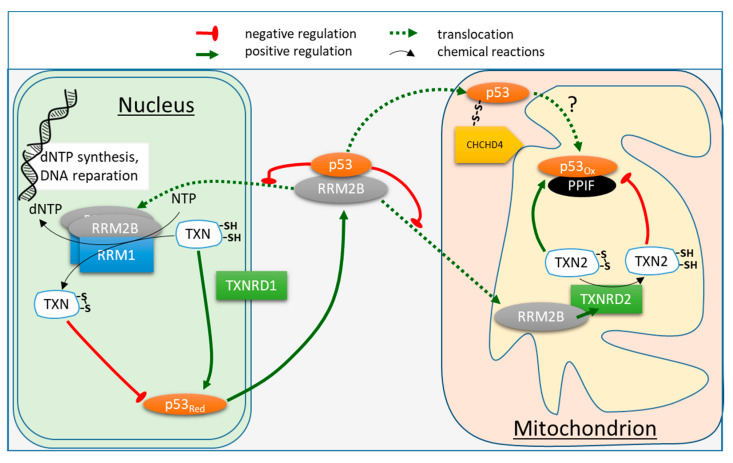
Interplay between the non-nuclear heterologous complexes of p53 and thiol-disulfide oxidoreductases. The names of cellular compartments are underlined. Cytosolic p53 sequesters RRM2B subunit of ribonucleotide reductase in cytosol. Under stress conditions, the p53-dependent expression of RRM2B (a p53 target gene) would cause accumulation of free RRM2B, which translocates to nucleus and mitochondria for the on-site synthesis of deoxyribonucleotides (dNTPs) as part of the DNA damage response program. The formation of active ribonucleotide reductase complex comprising RRM1 and RRM2B in nucleus is in accord with available experimental data [111,112,113,114]. Cytosolic thioredoxin (TNX) is a reducing substrate for the dNTPs synthesis on one hand, and a reductant of p53 disulfides on the other hand. The TNX shift to the disulfide form would decrease the dithiol form of p53 required for its transcriptional activity. The TNX shift to the reduced form, on the contrary, would increase the dithiol form of p53, thus increasing its transcriptional activity. P53 binds with far higher affinity to RRM2B promoter than to the proapoptotic gene promoters [45]. This hierarchy supports p53-dependent DNA repair without activating an apoptotic response. In the mitochondrial intermembrane space, p53 forms a mixed disulfide with CHCHD4, probably involved in p53 transport to mitochondrial matrix [16]. In mitochondrial matrix, RRM2B activates mitochondrial thioredoxin reductase (TXNRD2) upon their interaction [117]. The increased reduction of mitochondrial thioredoxin (TNX2) may not only stimulate the RRM2B-supported mitochondrial DNA repair, similar to that in nucleus, but also promote formation of the active reduced form of intramitochondrial p53. TNX2 and p53 compete for their complexes with PPIF [118], providing for the redox-dependent regulation of apoptosis. The p53-PPIF complex promotes apoptosis [54], the TNX2-PPIF complex promotes survival [118].

## Data Availability

Not applicable.

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
