# Peer review of "Regulation of p53 Function by Formation of Non-Nuclear Heterologous Protein Complexes"

_biomolecules, 2022, doi:10.3390/biom12020327_

Round 1

Reviewer 1 Report

The manuscript describes the scientific literature related to heterologous and non-nuclear interactions of p53. A very important topic that often receives poor attention. In my opinion the Review is comprehensive and well structured. The figures and tables are sufficiently informative. I suggest to the authors to improve these two minor aspects:
1- update the references because although some references are the orginal articles from which the discovery starts, often there are evolutions in recent literature that deserve to be cited.
2- implement the paragraph 5.2 Endoplasmic Reticulum with comments on what is reported and eventual mention of ER stress.

Reviewer 2 Report

In this review the authors summarized the p53 heterologous complexes that are able to fine tuning the settings-dependent p53 programs.

The review is interestinf and up-to date. The organization and the comprehension of the text could be improved. There are some minor points to amend.

-Abstract, lane 11: “A transcription factor p53 triggers either correction or cell death in metabolically stressed cells”. Correction of what? Please amend.

-I guess that you should add “DNA damage” as the main activator of p53 and not only “metabolic stress”. Please change.

-Introduction, lane 47: “In addition to nucleus, p53 is also found in other cellular compartments, such as mitochondria, endoplasmic reticulum and cytosol”, please add this reference: Nature. 2009 Apr 30; 458(7242): 1127.doi: 10.1038/nature07986

-In the Introduction the authors should add the information regarding the mutant p53.

-In paragraph 2 the reference to the figures in the text and the description of to the several panels in the figures should be more accurate for the understandable of the p53 structure. That is, instead of making a large figure the authors could split the figures in different parts following the description given in the text.

-As above, Figure 4 could be simplified.

-Figures are in general too complicated and could be simplified and insert in the text in a more accurate way, following the text.

Reviewer 3 Report

Overall, the review is quite complete and depicts the panel of effects produced by the protein-protein interactions of p53 and the cellular localization of these interactions. A few comments/suggestions, though, are listed below:

On page 5, line 150, the authors state that "As a result, loss-of-function due to the perturbed interaction with one partner is accompanied by gain-of-function due to the acquired ability to bind another partner.". this idea should be further explored throughout the review. What are the new interactions promoted by p53 mutations that generate gains-of-function? Whiche= gains-of-function are these? 

With the many papers published referring to p53 aggregation and coaggregation with other proteins, such as the paralogs p63 and p73, this reviewer suggests that this subject should be mentioned in this work, since it is an important part of the explanation of the pro-oncogenic effects of mutant p53.

A little comment on a paragraph: references are missing in pages 4-5, lines 119 (page 4) -151 (page 5).
